# Oxygen systems and quality of care for children with pneumonia, malaria and diarrhoea: Analysis of a stepped-wedge trial in Nigeria

Hamish R. Graham[1,2]*, Jaclyn Maher[3], Ayobami A. Bakare[2], Cattram D. Nguyen[3,4], Adejumoke I. Ayede[2,5], Oladapo B. Oyewole[2], Amy Gray[1], Rasa Izadnegahdar[6,7], Trevor Duke[1], Adegoke G. Falade[2,5]

1 Centre for International Child Health, The Royal Children's Hospital, MCRI, University of Melbourne, Parkville, Australia, 2 Department of Paediatrics, University College Hospital, Ibadan, Nigeria, 3 Department of Paediatrics, Royal Children's Hospital, University of Melbourne, Parkville, Australia, 4 Clinical Epidemiology and Biostatistics Unit, MCRI, Royal Children's Hospital, Parkville, Australia, 5 Department of Paediatrics, University of Ibadan, Ibadan, Nigeria, 6 Bill and Melinda Gates Foundation, Seattle, Washington, United States of America, 7 Department of Pediatrics, University of Washington, Seattle, Washington, United States of America

* hamish.graham@rch.org.au

**Data Availability Statement:** All dataset files are available from the OSF database: osf.io/pumdq.

## Abstract

### Objectives

To evaluate the effect of improved hospital oxygen systems on quality of care (QOC) for children with severe pneumonia, severe malaria, and diarrhoea with severe dehydration.

### Design

Stepped-wedge cluster randomised trial (unblinded), randomised at hospital-level.

### Setting

12 hospitals in south-west Nigeria.

### Participants

7,141 children (aged 28 days to 14 years) admitted with severe pneumonia, severe malaria or diarrhoea with severe dehydration between January 2014 and October 2017.

### Interventions

Phase 1 (pulse oximetry) introduced pulse oximetry for all admitted children. Phase 2 (full oxygen system) (i) standardised oxygen equipment package, (ii) clinical education and support, (iii) technical training and support, and (iv) infrastructure and systems support.

**Funding:** This work was funded by the Bill and Melinda Gates Foundation (OPP1123577), with support from the World Health Organisation, and state and federal health agencies. The funder was not involved in the writing of the manuscript or the decision to submit it for publication. The corresponding author had full access to the data and has final responsibility for the decision to submit for publication.

**Competing interests:** None declared.

**Abbreviations:** CRF, case report forms; LMIC, low- and middle-income countries; QOC, quality of care; $SPO_2$, peripheral blood oxygenation; The WHO, The World Health Organisation; U-5, under five years.

## Outcome measures

We used quantitative QOC scores evaluating *assessment*, *diagnosis*, *treatment*, and *monitoring* practices against World Health Organization and Nigerian standards. We evaluated mean differences in QOC scores between study periods (baseline, oximetry, full oxygen system), using mixed-effects linear regression.

## Results

7,141 eligible participants; 6,893 (96.5%) had adequate data for analysis. Mean paediatric QOC score (maximum 6) increased from 1.64 to 3.00 (adjusted mean difference 1.39; 95% CI 1.08–1.69, p<0.001) for severe pneumonia and 2.81 to 4.04 (aMD 1.53; 95% CI 1.23–1.83, p<0.001) for severe malaria, comparing the full intervention to baseline, but did not change for diarrhoea with severe dehydration (aMD -0.12; 95% CI -0.46–0.23, p = 0.501). After excluding practices directly related to pulse oximetry and oxygen, we found aMD 0.23 for severe pneumonia (95% CI -0.02–0.48, p = 0.072) and 0.65 for severe malaria (95% CI 0.41–0.89, p<0.001) comparing full intervention to baseline. Sub-analysis showed some improvements (and no deterioration) in care processes not directly related to oxygen or pulse oximetry.

## Conclusion

Improvements in hospital oxygen systems were associated with higher QOC scores, attributable to better use of pulse oximetry and oxygen as well as broader improvements in clinical care, with no negative distortions in care practices.

## Trial registration

ACTRN12617000341325

## Introduction

Effective oxygen systems are an important element of quality hospital care and are critical for achieving good clinical outcomes for patients with severe pneumonia [1, 2]. Improvements in hospital oxygen systems have been shown to reduce inpatient child mortality from pneumonia by 35–58%, yet oxygen access for patients remains poor in many low- and middle-income countries (LMICs) [3–8]. The COVID-19 pandemic has further highlighted the importance of improving hospital oxygen systems, and the fatal consequences of deficient systems, pushing oxygen access higher onto the global health agenda [9, 10].

Improvement in hospital oxygen systems requires attention to multiple domains which are relevant to the overall quality of care in health facilities, including *structures* (e.g. equipment, policy, financing), care *processes*, (e.g. detections of hypoxaemia and appropriate use of oxygen) and therefore *outcomes* (e.g. case fatality) [4, 11, 12]. This includes activities to increase availability of oxygen (e.g. oxygen equipment procurement, maintenance and repair), improve clinical use of oxygen (e.g. routine pulse oximetry and clinical guidelines) and connect relevant staff cadres (e.g. doctors, nurses, technicians, managers).

Findings from previous oxygen improvement programs suggest that clinical impacts are more significant if oxygen is approached as part of a more comprehensive quality

improvement program [3, 6, 8]. Authors postulated that improvements in oxygen systems may have contributed to broader improvements in *structures* and *processes* of care, thereby enabling the oxygen improvement program to achieve more substantial reductions in mortality, than would have been possible with a narrower approach to oxygen systems [3, 6]. However, it is also possible that broad-reaching interventions to improve oxygen systems could have unintended negative effects on quality of care.

Nigeria is a populous lower middle-income country in sub-Saharan Africa with a mortality rate in children under five years of age (U-5) of 120 deaths per 1000 live births—the second highest rate globally in 2018 [13, 14]. Nigeria contributes one-third of U-5 malaria deaths and one-sixth of U-5 pneumonia deaths globally [15]. Hospital oxygen systems in Nigeria are generally weak, with recent studies showing that only one in five children with evidence of low blood oxygen (hypoxaemia) received oxygen therapy, despite hypoxaemia being common (10% of hospitalised children had a blood oxygen saturation <90%) and fatal [16, 17].

The Nigeria Oxygen Implementation program implemented and evaluated improved oxygen systems in 12 secondary health facilities in south-west Nigeria. Results are published elsewhere [5, 16–22]. We found that the introduction of pulse oximetry and improved oxygen systems contributed to marked improvements in pulse oximetry and oxygen practices and was associated with reduced risk of death for children admitted with pneumonia, but not other illnesses [5]. Importantly, we observed the greatest impact on pneumonia mortality with the introduction of pulse oximetry (prior to implementing the full oxygen system) and received qualitative feedback that pulse oximetry may have had broader positive effects on other care practices [5, 19]. However, we do not know the extent to which our multi-faceted oxygen improvement program may have impacted (positively or negatively) on broader quality of care.

This paper seeks to evaluate the effect of a multi-faceted oxygen system intervention on the quality of care provided for children (aged 1 month to 14 years) admitted to hospital with severe pneumonia, severe malaria and diarrhoea with severe dehydration, and explore potential positive and negative effects beyond oxygen-related aspects of care.

## Methods

### Ethics statement

We obtained ethics approval from the University of Melbourne (1543797.1) and University of Ibadan/University College Hospital Ethics Committee, Ibadan, Nigeria (UI/EC/16/0413). Patients whose records were accessed during this study sought treatment between November 2013 and October 2017. Both retrospective and prospective data collection occurred between September 2015 and December 2017. Data was fully anonymised upon access by the authors. The ethics committees did not require us to have individual patient consent.

### Study design

This study was a stepped-wedge cluster randomised trial conducted in 12 Nigerian hospitals. Following a pre-specified timetable, we introduced pulse oximetry to all 12 hospitals in November 2015, followed by stepped introduction of the full oxygen system—clusters of three hospitals, every four months between March 2016 and March 2017, with gradual withdrawal of support to encourage full site self-sustainability by the end of 2019 (Fig 1, S1 Fig). Risk of contamination was minimised by the geographic separation of hospitals. Full study details are available elsewhere [5, 18].

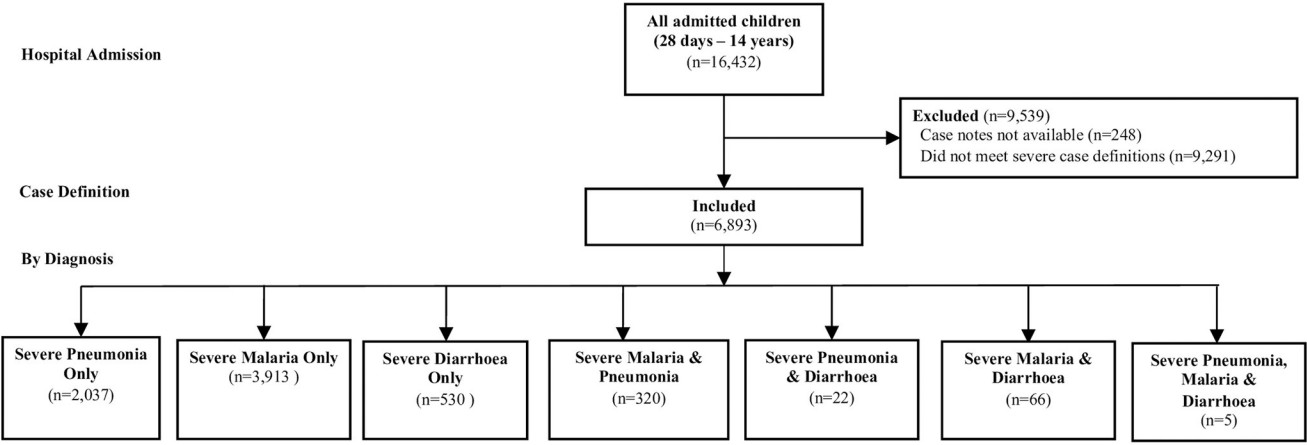

**Fig 1. Participant inclusion in analysis, by diagnosis (all steps).**

## Sample

We selected hospitals in collaboration with state health authorities, identifying secondary-level health facilities that admitted children (see study protocol for detail: DOI 10.1186/s13063-017-2241-8) [18]. The 12 hospitals included a mix of government and mission (private non-profit) hospitals across four states in south-west Nigeria, including three hospitals that focussed on paediatric care (details in S1 File). Hospital representatives provided consent for participation prior to randomisation and we did not require individual patient/carer consent. We collected data on all children (aged below 15 years), admitted to participating hospitals during the study period.

For this analysis, we included participants aged 28 days to 14 years who met World Health Organization (WHO) definitions for severe pneumonia, severe malaria or diarrhoea with severe dehydration (S1 Table). We accepted multiple diagnoses (Fig 1). For outcome analysis we excluded participants admitted in the two-weeks at the beginning of each crossover period ("wash-out period"), to implement the intervention and avoid contamination.

## Randomisation

We randomly allocated hospitals to receive the full oxygen system at pre-specified dates using a computer-based random sequence generator. Concealment of the intervention or its start date was not possible as implementation required active participation from hospital staff.

## Intervention

The intervention involved two phases. Phase 1 (pulse oximetry) introduced pulse oximetry for all admitted children and included provision of pulse oximeters and basic training. Phase 2 (full oxygen system) involved (i) a standardised oxygen equipment package, (ii) clinical education and support, (iii) technical training and support, and (iv) infrastructure and systems support. These all focussed on pulse oximetry and oxygen therapy, without substantially addressing broader clinical or technical topics. We designed the intervention based on experience from previous oxygen projects [18, 23, 24], WHO guidelines [25–27], and were informed by educational, behaviour change and implementation theories and strategies [28–32]. Further information on local adaptation of the intervention is published elsewhere [5, 18].

## Public & patient involvement

We did not involve patients directly in the planning or conduct of this study. We involved members of the hospital teams, government, and the WHO throughout the planning, implementation, and evaluation of the program, and supported public events at facilities to share results.

## Procedures

Trained nurse data collectors at each facility extracted data from case notes (made by clerking doctors as per usual procedure) using standardised Case Report Forms (CRFs) after identifying eligible patients from the ward admission book. The CRFs included data on i) demographics, ii) initial assessment, iii) oxygen therapy, iv) findings & case definitions and v) processes of care for each of pneumonia, malaria and diarrhoea. We collected data retrospectively for two years pre-intervention (January 2014 to October 2015), and prospectively between November 2015 and October 2017.

During the pre-intervention period, hospitals provided care as usual, including their existing supplies of oxygen. During the pulse oximetry period, hospitals continued usual care with the additional availability of project-supplied pulse oximeters. Following the full oxygen system intervention (starting March 2016, July 2016, November 2016, or March 2017), hospitals provided clinical care using the new oxygen systems, clinical guidelines and maintenance protocols.

## Program theory

We hypothesized that the intervention would improve (i) reliability of oxygen availability, and (ii) healthcare worker use of oxygen, thereby increasing the proportion of hypoxaemic patients receiving oxygen, contributing towards reduced inpatient mortality. We expected the benefit to be greatest for children with pneumonia, for whom hypoxaemia is very common. We acknowledged the possibility that a multi-faceted oxygen intervention may affect quality of care more broadly, both positively (e.g. improved monitoring of sick patients) and negatively (e.g. over-reliance on oxygen to the exclusion of other therapies).

## Outcomes

This study focused on pre-specified process outcomes. The primary outcome for this study was the mean change in quality of care (QOC) composite scores for children admitted with severe pneumonia, severe malaria and diarrhoea with severe dehydration.

We used data extracted from the case notes to classify diagnoses according to the presence of syndrome defining clinical signs, using standard WHO case definitions, (S1 Table) and generate disease specific QOC scores (S2 Table). The QOC scores evaluated assessment, diagnosis, treatment, and monitoring against WHO and Nigerian standard guidelines for each illness [27, 33]. The content and weighting of each domain reflected the key steps in management: assessment of primary and secondary signs/symptoms; diagnostic classification; treatment with primary and supportive therapies; and monitoring of vital signs. We designed this score based on previous work in Laos [34], and it is conceptually similar to the recently validated Paediatric Admission Quality of Care (PAQC) score from Kenya [35, 36]. Where clinical signs, clinical investigations or treatment were not documented, we assumed they were not present or done. We conducted quality checks on the data extraction after initial training, and periodically throughout the program, to satisfy ourselves that the data accurately represented what was documented in patient files.

We assigned disease-specific scores at the patient-level. Where children had more than one of the three diagnoses, they were assigned a score for each illness. We report both the disease specific QOC score totals and individual component scores (assessment, diagnosis, management, monitoring) to show how each domain contributed to the overall score. To evaluate the intervention's effect on non-oxygen-related aspects of care, we excluded score components related directly to pulse oximetry and oxygen therapy ("modified score").

## Statistical analysis

We performed all analyses using Stata 15.1 (Statacorp, Texas, USA). We described patient characteristics by study period and summarised data using frequencies and percentages for categorical variables, and medians and interquartile ranges for continuous variables. We represented the crude means both in tabular form and visually in heat-map tables, by hospital and time period.

We reported intervention effects as mean differences in total QOC scores, adjusted for clustering and time using mixed-effects linear regression models. Our analysis model included fixed effects for intervention and time (four-month periods) and random effects for cluster (hospital), and was fitted to individual-level continuous outcome data [37, 38]. We found evidence of hospital-time interaction (by comparing the simple model with a model that included a variance component for random by-time effects, using a likelihood ratio test), and adjusted models accordingly. Primary analysis included the pre-intervention data, pulse oximetry period, and the full oxygen system (steps 1–12) to provide estimates of effect of pulse oximetry and full oxygen system periods against baseline. We also performed a restricted analysis (steps 7–12) comparing the stepped introduction of the full oxygen system to the pulse oximetry period alone (S3 Table).

# Results

During the study period, 16,432 children were admitted to the participating hospitals, including 6,893 who met criteria for inclusion and for whom adequate data was available (Fig 1). Individual hospitals contributed between 113 and 1310 participants each, of whom almost two-thirds had severe malaria, approximately one-third had severe pneumonia, and <10% had diarrhoea with severe dehydration (Table 1). Further information, by illness, can be found in S4 Table.

The distribution of patient diagnoses varied slightly over the study periods. Patients were slightly older and tended to present with more severe illness (e.g. higher proportion with altered conscious state or hypoxaemia), during the pulse oximetry and full oxygen system periods.

## Intervention effects

Tables 2 and 3 show the crude and adjusted mean QOC scores for children with severe pneumonia, severe malaria and diarrhoea with severe dehydration. Fig 2 demonstrates the trend in QOC score means over time, by hospital for each illness. Additional detail on score components contained in S5 Table.

**Severe pneumonia.**　Crude mean QOC scores for children with severe pneumonia increased almost two-fold, from 1.64 to 3.00 (out of a maximum 6 points), between the pre-intervention and full oxygen system periods (Table 2). Heat map tables showed that this change was largely achieved during the pulse oximetry period, with considerable variability between hospitals (Fig 2). Adjusted analysis (Table 3) showed that mean QOC score totals for children with severe pneumonia increased by 0.66 points (95% CI 0.22–1.09, p = 0.003) in the

**Table 1. Population characteristics of participants, by study period.**

| | Pre-intervention period (n = 3 128) | Pulse oximetry only period (n = 1 564) | Full oxygen system period (n = 2 161) |
|---|---|---|---|
| **Infant (1–11 months)** | 876 (28.0%) | 474 (30.2%) | 576 (26.7%) |
| **Young child (1–4 years)** | 1881 (60.1%) | 859 (54.9%) | 1183 (54.7%) |
| **Older child (5–14 years)** | 371 (11.9%) | 232 (14.8%) | 402 (18.6%) |
| **Age, months median (IQR)** | 21.0 (10.4–36) | 19.1 (10.0–39.0) | 24.0 (11.0–48.0) |
| **Sex, % female** | 45.30% | 43.90% | 44.70% |
| **Hospital type, % government** | 56.70% | 64.00% | 56.50% |
| **Length of stay, days median (IQR)** | 4 (2–5) | 4 (2–5) | 4 (2–6) |
| **Child diagnoses and presenting signs[2]** | | | |
| **Severe pneumonia** | 1039 (33.0%) | 571 (36.4%) | 774 (35.5%) |
| **Severe malaria** | 2005 (63.7%) | 921 (58.7%) | 1378 (63.2%) |
| **Severe diarrhoea** | 292 (9.3%) | 171 (10.9%) | 160 (7.3%) |
| **Malnutrition** | 56 (1.8%) | 43 (2.8%) | 42 (1.9%) |
| **HIV** | 2 (0.06%) | 5 (0.32%) | 4 (0.18%) |
| **Severe respiratory distress** | 851 (27.1%) | 455 (29.0%) | 523 (24.0%) |
| **Packed cell volume below 15%** | 672 (23.0%) | 273 (18.7%) | 508 (24.8%) |
| **Signs of severe dehydration** | 322 (10.2%) | 194 (12.4%) | 183 (8.4%) |
| **Central cyanosis** | 18 (0.57%) | 14 (0.89%) | 40 (1.84%) |
| **Hypoxaemia (SpO$_2$<90%)[4]** | 32 (14.2%) | 228 (23.0%) | 497 (24%) |
| **Severe hypoxaemia (SpO$_2$<80%)[4]** | 15 (6.7%) | 90 (9.1%) | 240 (11.6%) |
| **Signs of shock** | 15 (0.48%) | 12 (0.77%) | 15 (0.69%) |
| **Unable to feed** | 791 (25.1%) | 371 (23.7%) | 551 (25.3%) |
| **Convulsions** | *1194* (50.5%) | 499 (46.5%) | 696 (49.9%) |
| **Confusion or lethargy** | 713 (35.3%) | 470 (49.2%) | 436 (35.2%) |
| **Coma or barely conscious** | 403 (12.8%) | 183 (11.7%) | 408 (18.7%) |
| **Any WHO Emergency Sign** | 2605 (82.8%) | 1371 (87.4%) | 1860 (85.7%) |

Data are *n* (%) unless otherwise indicated. Severe pneumonia, severe malaria and diarrhoea with severe dehydration as per our case definitions. Other diagnoses as per admission diagnosis.

Abbreviations: SD = standard deviation, IQR = interquartile range, SpO$_2$ = peripheral oxygen saturation

[0]Pearson Chi$^2$ unless otherwise specified

[1]K-sample equality of medians test and Wilks' lambda test of means

[2]Multiple diagnoses permitted

[3] Fisher's exact test

[4]Limited hypoxaemia data available in pre-intervention period, mostly from a single hospital.

pulse oximetry period and 1.39 points (95% CI 1.08–1.69, p<0.001) in the full oxygen period, compared to baseline.

Analysis of the score components showed that assessment, treatment, and monitoring scores for children with severe pneumonia increased across the intervention periods, compared to baseline, with no change in diagnosis scores (Table 3). Assessment scores increased by 0.41 points (out of a maximum 2 points) (95% CI 0.13–0.69, p = 0.004) in the pulse oximetry period and 0.74 points (95% CI 0.54–0.93, p<0.001) in the full oxygen period, compared to baseline. Diagnosis scores (max 1) showed no evidence of change in either the pulse oximetry (aMD = -0.04; -0.12–0.04, p = 0.342) or the full oxygen system periods (aMD = -0.02; -0.08–0.04, p = 0.559), when compared to baseline. Treatment scores (max 2) changed minimally in the pulse oximetry period (aMD = 0.15; 95% CI -0.01–0.31, p = 0.072), but were higher in the full oxygen period (aMD = 0.32, 95% CI 0.20–0.44, p<0.001), compared to baseline.

**Table 2. Mean QOC scores for children with severe pneumonia, malaria, and diarrhoea, showing crude changes over time and by score component (assessment, diagnosis, treatment, monitoring).**

| | Mean QOC score (SD) | | | | | Mean modified QOC score (SD)* | | | |
|---|---|---|---|---|---|---|---|---|---|
| | Assessment | Diagnosis | Treatment | Monitoring | TOTAL | Modified Assessment | Modified Treatment | Modified Monitoring | Modified TOTAL |
| Severe pneumonia QOC scores | | | | | | | | | |
| - Pre-intervention | 0.47 (0.57) | 0.32 (0.24) | 0.42 (0.54) | 0.43 (0.50) | **1.64 (0.89)** | 1.32 (0.57) | 0.69 (0.53) | 0.84 (0.37) | **3.16 (0.90)** |
| - Pulse oximetry | 1.02 (0.67) | 0.30 (0.25) | 0.56 (0.54) | 0.65 (0.48) | **2.53 (1.07)** | 1.24 (0.53) | 0.74 (0.49) | 0.81 (0.39) | **3.09 (0.85)** |
| - Full O2 system | 1.24 (0.55) | 0.29 (0.25) | 0.67 (0.55) | 0.80 (0.36) | **3.00 (0.94)** | 1.29 (0.51) | 0.73 (0.52) | 0.90 (0.30) | **3.21 (0.82)** |
| Severe malaria QOC scores | | | | | | | | | |
| - Pre-intervention | 0.27 (0.52) | 0.73 (0.37) | 1.07 (0.72) | 0.73 (0.44) | **2.81 (1.16)** | 1.07 (0.51) | 1.31 (0.69) | 0.84 (0.36) | **3.95 (1.11)** |
| - Pulse oximetry | 0.70 (0.64) | 0.74 (0.37) | 1.29 (0.42) | 0.77 (0.42) | **3.51 (1.12)** | 1.09 (0.43) | 1.49 (0.62) | 0.83 (0.38) | **4.14 (0.96)** |
| - Full O2 system | 1.04 (0.42) | 0.74 (0.38) | 1.44 (0.62) | 0.80 (0.39) | **4.04 (0.95)** | 1.08 (0.38) | 1.57 (0.60) | 0.88 (0.32) | **4.28 (0.90)** |
| Diarrhoea with severe dehydration QOC scores | | | | | | | | | |
| - Pre-intervention | 1.30 (0.61) | 0.86 (0.25) | 0.28 (0.50) | 0.83 (0.38) | **3.28 (0.94)** | - | - | - | - |
| - Pulse oximetry | 1.21 (0.59) | 0.86 (0.28) | 0.22 (0.47) | 0.78 (0.42) | **3.07 (0.98)** | - | - | - | - |
| - Full O2 system | 1.34 (0.58) | 0.89 (0.21) | 0.33 (0.51) | 0.88 (0.32) | **3.44 (0.96)** | - | - | - | - |

QOC = quality of care; SD = standard deviation.

Scoring summed to a maximum of 6 points in total (2-point assessment, 1-point diagnosis, 2-points treatment, 1-point monitoring)

*Modified score = modified to exclude pulse oximetry and oxygen therapy components from scores.

Monitoring scores (max 1) also changed minimally in the pulse oximetry period (aMD = 0.19; 0.001–0.39, p = 0.049) but were higher in the full oxygen period (aMD = 0.37, 95% CI 0.24–0.51, p<0.000), compared to baseline.

Modified QOC score totals (omitting score items directly related to pulse oximetry and oxygen therapy) for children with severe pneumonia showed lesser change over the study period (Tables 2 and 3), suggesting that most of the improvements in QOC scores were directly related to pulse oximetry and oxygen therapy practices. Crude mean modified QOC scores for children with severe pneumonia were 3.16 (SD 0.90) in the pre-intervention period, 3.09 (SD 0.85) in the pulse oximetry period and 3.21 (SD 0.82) in the full oxygen system period (max 6) (Table 2). Adjusted analysis showed that the mean difference in modified QOC score was -0.17 points in the pulse oximetry period (-0.53–0.19, p = 0.351), and 0.23 points in the full oxygen system period (-0.02–0.48, p = 0.072), compared to baseline. Component analysis suggested minimal effect on modified assessment (aMD -0.05; -0.22,0.13, p = 0.601), treatment (aMD 0.17; 0.05–0.29, p = 0.006) and monitoring (aMD = 0.13; 0.02–0.24, p = 0.016) scores in the full oxygen system period, compared to baseline.

Detailed sub-analysis of score components for children with severe pneumonia revealed substantial improvements in pulse oximetry coverage (increasing from 6% to 95% between pre-intervention and full oxygen system periods) and oxygen for children with hypoxaemia on admission (50% to 83%) (S5 Table). However, there was relatively poor documentation of feeding status (~50%), diagnostic classification (one-third correctly classified as severe

**Table 3. Effect of the intervention on non-modified and modified mean QOC score totals and components, comparing the pulse oximetry and full oxygen system periods to baseline (see S3 Table for restricted analysis comparing full oxygen system period to pulse oximetry period).**

| | Severe pneumonia | | | | Severe malaria | | | | Diarrhoea with severe dehydration | |
|---|---|---|---|---|---|---|---|---|---|---|
| | Non-modified[1] | | Modified[2] | | Non-modified[1] | | Modified[2] | | Non-modified[1] | |
| | aMD (95% CI) | p-value | aMD (95% CI) | p-value | aMD (95% CI) | p-value | aMD (95% CI) | p-value | aMD (95% CI) | p-value |
| **Score total [3]** | | | | | | | | | | |
| Pulse oximetry | 0.66 (0.22, 1.09) | 0.003 | -0.17 (-0.53, 0.19) | 0.351 | 1.26 (0.85, 1.68) | <0.001 | 0.68 (0.35, 1.00) | <0.001 | -0.29 (-0.78, 0.21) | 0.259 |
| Full O2 system | 1.39 (1.08, 1.69) | <0.001 | 0.23 (-0.02, 0.48) | 0.072 | 1.53 (1.23, 1.83) | <0.001 | 0.65 (0.41, 0.89) | <0.001 | -0.12 (-0.46, 0.23) | 0.501 |
| **Assessment component[3]** | | | | | | | | | | |
| Pulse oximetry | 0.41 (0.13, 0.69) | 0.004 | -0.19 (-0.44, 0.06) | 0.138 | 0.43 (0.20, 0.66) | <0.001 | -0.03 (-0.19, 0.13) | 0.705 | -0.25 (-0.60, 0.10) | 0.163 |
| Full O2 system | 0.74 (0.54, 0.93) | <0.001 | -0.05 (-0.22, 0.13) | 0.601 | 0.67 (0.51, 0.83) | <0.001 | -0.06 (-0.18, 0.06) | 0.303 | -0.20 (-0.44, 0.04) | 0.097 |
| **Diagnosis component[3]** | | | | | | | | | | |
| Pulse oximetry | -0.04 (-0.12, 0.04) | 0.342 | - | - | 0.15 (0.06, 0.25) | 0.002 | - | - | -0.03 (-0.17, 0.11) | 0.683 |
| Full O2 system | -0.02 (-0.08, 0.04) | 0.559 | - | - | 0.12 (0.04, 0.10) | 0.002 | - | - | -0.05 (-0.15, 0.05) | 0.292 |
| **Treatment component[3]** | | | | | | | | | | |
| Pulse oximetry | 0.15 (-0.013, 0.31) | 0.072 | 0.06 (-0.11, 0.22) | 0.511 | 0.55 (0.33, 0.78) | <0.001 | 0.47 (0.27, 0.68) | <0.001 | -0.01 (-0.27, 0.25) | 0.942 |
| Full O2 system | 0.32 (0.20, 0.44) | <0.001 | 0.17 (0.05, 0.29) | 0.006 | 0.56 (0.39, 0.73) | <0.001 | 0.46 (0.31, 0.61) | <0.001 | 0.10 (-0.08, 0.28) | 0.264 |
| **Monitoring component[3]** | | | | | | | | | | |
| Pulse oximetry | 0.19 (0.001, 0.39) | 0.049 | 0.04 (-0.11, 0.20) | 0.601 | 0.12 (-0.04, 0.27) | 0.139 | 0.07 (-0.06, 0.20) | 0.832 | -0.11 (-0.34, 0.11) | 0.317 |
| Full O2 system | 0.37 (0.24, 0.51) | <0.001 | 0.13 (0.02, 0.24) | 0.016 | 0.18 (0.07, 0.29) | 0.002 | 0.13 (0.04, 0.22) | 0.007 | -0.002 (-0.16, 0.15) | 0.976 |

Acronyms: aMD = adjusted mean difference

[1] Non-modified QOC scores include features related to pulse oximetry and oxygen system use

[2] Modified QOC scores exclude features related to pulse oximetry and oxygen system use

[3] Total = mean out of 6 total points, Assessment = total out of 2 points, Diagnosis = total out of 1 point, Treatment = total out of 2 points, Monitor = total out of 1 point

pneumonia) and antibiotic prescribing (<6% prescribed appropriate first-line antibiotics), with minimal change over the study period.

**Severe malaria.** Crude mean QOC scores for children with severe malaria increased from 2.81 to 4.04 (out of a maximum 6 points), between the pre-intervention and full oxygen system periods (Table 2). Heat map tables (Fig 2) show a gradual trend towards improved quality of care for children admitted with severe malaria across each of the study periods, with some variability over time and between hospitals. Adjusted analysis (Table 3) showed that mean QOC non-modified score totals for children with severe malaria increased in the pulse oximetry period by 1.26 points (95% CI 0.85–1.68, p<0.001) and the full oxygen system period by 1.53 points (95% CI 1.23–1.83, p<0.001), compared to baseline.

Analysis of the score components showed that assessment, diagnosis, treatment and monitoring scores for severe malaria all increased in the intervention periods, compared to baseline. Assessment scores (out of a maximum 2 points) were increased in the pulse oximetry period by 0.43 points (95% CI 0.20–0.67, p = <0.001) and in the full oxygen period by 0.67 points (95% CI 0.51–0.83, p<0.000) compared to baseline. Diagnosis scores (max 1) were increased in the pulse oximetry period by 0.15 points (95% CI 0.0.06–0.25, p = 0.002) and full oxygen period by 0.12 (95% CI 0.04–0.10, p = 0.002) compared to baseline. Treatment scores (max 2) were increased in the pulse oximetry period by 0.55 points (95% CI 0.33–0.78, p<0.001) and full oxygen period by 0.56 points (95% CI 0.39–0.73, p<0.001), compared to baseline. Monitoring scores (max 1) showed no change in the pulse oximetry period (aMD = 0.12, 95% CI -0.04–0.27, p = 0.139) but were higher in the full oxygen period (aMD = 0.18, 95% CI 0.07, 0.29, p = 0.002), compared to baseline.

**Steps (4-month periods)**

**A. Severe pneumonia QOC score mean (n=2384)**

| | 1 | 2 | 3 | 4 | 5 | 6 | 7 | 8 | 9 | 10 | 11 | 12 |
|---|---|---|---|---|---|---|---|---|---|---|---|---|
| 1 | 1.62 | 1.69 | 1.64 | 1.5 | 1.64 | 1.59 | 2.73 | 3.12 | 3.2 | 2.95 | 2.87 | 3.06 |
| 2 | 1.45 | 1.24 | 1.17 | 1.13 | 1.07 | 1.21 | 1.95 | 2.7 | 2.58 | 2.74 | 2.48 | 2.5 |
| 3 | 1.5 | 1.56 | 1.42 | 1.18 | 1.1 | 1.61 | 2.69 | 3.94 | 3.53 | 2.86 | 3.38 | 3.28 |
| 4 | 1.36 | 1.29 | 1.55 | 1.67 | 1.5 | 1.77 | 2.57 | 2.82 | 3.25 | 3.32 | 2.81 | 2.85 |
| 5 | 3.43 | 4.04 | 3.35 | 3.61 | 2.63 | 3.75 | 3.95 | 2.91 | 3.38 | 3.5 | 3.5 | 3.31 |
| 6 | 1.62 | 1.63 | 2.1 | 1.75 | 2 | 2.11 | 2.8 | 3.38 | 3.83 | 3.59 | 3.75 | 3.93 |
| 7 | 1.08 | 1.13 | 1.5 | 1.28 | 1.41 | 1.64 | 2.21 | 2.23 | 2.05 | 2.49 | 2.34 | 3.05 |
| 8 | 1.5 | 1.71 | | 1.5 | | 2.25 | 3.5 | 2.64 | 1 | 2.67 | 3 | 2 |
| 9 | 1.88 | 2.14 | 1.92 | 2.1 | 1.92 | 1.89 | 3.22 | 2.67 | 2.33 | 2.29 | 2.65 | 2.86 |
| 10 | 1.63 | 1.9 | 1.38 | 1.81 | 1.58 | 1.63 | 2.54 | 2.83 | 2.51 | 2.33 | 2.8 | 2.64 |
| 11 | 2 | 1.4 | 1.1 | 0.9 | 2 | 2.88 | 3.1 | 3.6 | 2.33 | 3 | 3.75 | 3.9 |
| 12 | 1.5 | 1.5 | | | 2.5 | 2.17 | 2 | 1.63 | 1 | 2.25 | 3.5 | 2.5 |

**B. Severe malaria QOC score mean (n=4,304)**

| | 1 | 2 | 3 | 4 | 5 | 6 | 7 | 8 | 9 | 10 | 11 | 12 |
|---|---|---|---|---|---|---|---|---|---|---|---|---|
| 1 | 3.05 | 2.77 | 2.57 | 2.68 | 2.64 | 3.24 | 3.61 | 3.83 | 3.85 | 4.04 | 4.33 | 4.11 |
| 2 | 2.28 | 2.88 | 3.07 | 2.92 | 2.71 | 3.11 | 3.3 | 4.17 | 4.3 | 3.98 | 3.87 | 4.09 |
| 3 | 2.5 | 1.84 | 2.21 | 1.75 | 2.3 | 2.38 | 2.67 | 3.55 | 3.56 | 3.5 | 3.86 | 3.7 |
| 4 | 2.1 | 2.3 | 2.43 | 1.82 | 2.62 | 2.84 | 3.46 | 4 | 4 | 3.91 | 4.75 | 3.9 |
| 5 | 5.17 | 4.61 | 4.77 | 4.78 | 5.17 | 5.42 | 4.86 | 4.6 | 4.5 | 4.62 | 4.66 | 4.45 |
| 6 | 2.5 | 2.72 | 2.5 | 3.18 | 3.32 | 3.31 | 3.52 | 4.69 | 4.57 | 4.31 | 4.42 | 4.59 |
| 7 | 1.42 | 1.91 | 1.88 | 2.5 | 2.63 | 2.85 | 3.04 | 2.92 | 3.17 | 3.75 | 3.69 | 3.66 |
| 8 | | 2.09 | 1.75 | | 2.33 | 2.33 | 2 | 3.33 | 2.75 | 3.13 | 3.28 | 2.96 |
| 9 | 3.56 | 3.55 | 3.78 | 4.25 | 3.32 | 3.2 | 3.68 | 3.7 | 2.26 | 3.66 | 4.13 | 4.1 |
| 10 | 2.3 | 2.39 | 2.17 | 2.6 | 2.73 | 2.58 | 3.23 | 4.2 | 3.95 | 3.93 | 4.07 | 3.86 |
| 11 | 1.68 | 2 | 2.56 | 2.68 | 2.7 | 2.89 | 4.21 | 4.15 | 4.09 | 3.96 | 4.25 | 4.29 |
| 12 | 2 | 2.36 | 1 | 2.17 | 2.5 | 2.46 | 3 | 3.08 | 2.65 | 3.03 | 3.68 | 3.72 |

**C. Diarrhoea with severe dehydration QOC score mean (n=623)**

| | 1 | 2 | 3 | 4 | 5 | 6 | 7 | 8 | 9 | 10 | 11 | 12 |
|---|---|---|---|---|---|---|---|---|---|---|---|---|
| 1 | 3.88 | 3.77 | 3.42 | 3.31 | 3.25 | 4.5 | 3.75 | 3.89 | 3.72 | 3.67 | 4.04 | 3.44 |
| 2 | 2.95 | 3.41 | | | 4 | 4 | 3 | 3.75 | 3 | 2.9 | 3 | 3 |
| 3 | | 2.75 | 3.25 | 2.5 | 4 | 3 | 3 | | 3 | 3.25 | 3.75 | 3 |
| 4 | 3.43 | 4.5 | 3.5 | 3.5 | 3.5 | 3.67 | 3.75 | 3 | | 3 | 4.5 | 3.67 |
| 5 | 4.06 | 4 | 3.5 | 4.17 | | 4 | 3.88 | 3.5 | 4.5 | 4.12 | 4 | 3.5 |
| 6 | 3.5 | 3.36 | 3.4 | 3.5 | 3.45 | 4 | 4.04 | 4 | 3.88 | 3.83 | 3.8 | |
| 7 | 2.69 | 1.94 | | 3.04 | 3.13 | 3.3 | 2.94 | 2.28 | 2.86 | 2.31 | 2.44 | 3.25 |
| 8 | | | | | | | 2.5 | 3 | | | 2 | 2.25 |
| 9 | 3.67 | 4 | 4 | 4 | 4.25 | 3.67 | 3.63 | 2 | 4 | 2.5 | 3.5 | 2.25 |
| 10 | 2.93 | 2.71 | 2.45 | 2.75 | 2.67 | 2.71 | 2.88 | 3.5 | 3.5 | 2.83 | 4 | 2 |
| 11 | 2.75 | 2.67 | 2.67 | 2.64 | | | 2.67 | | | 3.4 | 3 | 3.8 |
| 12 | 2.5 | | | | | | | | 2.5 | 2.5 | 3 | 2.67 |

**Fig 2. Crude QOC score total means (maximum six points) for severe pneumonia, severe malaria and diarrhoea with severe dehydration, by hospital and step.** Each cell contains the crude mean QOC score for a hospital during that step. For each of the three panels in this figure, the 12 hospitals are represented on the y-axis, whereas the steps of the trial are represented on the x-axis. The colour gradient extends from red (lowest score) through yellow to green (highest scores), providing a visual representation of change in scores over time. Blue spark-lines show the hospital-specific trend for easy reference. When the denominator to compute the cell rate is 0, cells are coloured in white. Data from November/December 2013 not available in step 1.

Modified QOC score totals (omitting score items directly related to pulse oximetry and oxygen therapy) for children with severe malaria showed positive change over the study period (Table 3). Crude mean modified QOC scores for children with severe malaria were 3.95 (SD 1.11) in the pre-intervention period, 4.14 (SD 0.96) in the pulse oximetry period, and 4.28 (SD 0.90) in the full oxygen system period (max 6). Adjusted analysis of modified QOC scores showed that scores were higher in the pulse oximetry period (aMD = 0.68, 95% CI 0.35–0.89, p<0.001) and full oxygen period (aMD = 0.65, 95% CI 0.41–0.89, p<0.001), compared to baseline, with all the improvement realised during the pulse oximetry period. Component analysis suggested no effect on modified assessment scores (aMD = -0.06, 95% CI = -0.18–0.06, p = 0.303), but higher modified treatment (aMD = 0.47, 95% CI 0.27–0.68, p<0.001) and monitoring scores (aMD = 0.13; 0.02–0.24, p = 0.016) in the full oxygen system period, compared to baseline.

Detailed sub-analysis of score components for children with severe malaria revealed substantial improvements in pulse oximetry coverage (increasing from 8% to 95% between pre-intervention and full oxygen system periods) and oxygen for children with hypoxaemia on admission (15% to 46%) (S5 Table). However, there was relatively poor documentation of feeding status or blood sugar level on admission (~50% coverage).

**Diarrhoea with severe dehydration.** Crude mean QOC scores for children changed minimally, between the pre-intervention and full oxygen system periods (Table 2), with no obvious

trends evident in heat map tables (Fig 2). Adjusted analysis (Table 3) showed that mean QOC scores did not change in either the pulse oximetry period (MD = -0.29, 95% CI -0.78, 0.21, p = 0.259) or the full oxygen system period (MD = -0.12, 95% CI -0.46, 0.23, p = 0.501), compared to baseline. Component analysis showed no change in any of assessment (aMD = -0.20, 95% CI -0.44–0.04, p = 0.097), diagnosis (aMD = -0.05, 95% CI -0.15–0.05, p = 0.292), treatment (aMD = 0.10, 95% CI -0.08–0.28, p = 0.264) or monitoring scores (aMD = -0.002, 95% CI -0.16–0.15, p = 0.976), compared to baseline for diarrhoea with severe dehydration.

Detailed sub-analysis of score components for children with severe diarrhoeal disease revealed relatively poor documentation of feeding status on admission (~50% coverage), fluid prescribing (<15% prescribed appropriate rehydration fluids), and antibiotic prescribing (<20% appropriate), with little change over the study periods (S5 Table).

## Discussion

This study reports results from the Nigeria Oxygen Implementation program regarding the impact of a multifaceted oxygen system intervention on paediatric care practices for children with severe pneumonia, severe malaria and diarrhoea with severe dehydration. We found an improvement in QOC score totals following the introduction of pulse oximetry and improved oxygen systems, for both severe pneumonia and severe malaria, compared to usual care at baseline. These changes were smaller, but still evident, after excluding QOC score features that were directly related to pulse oximetry and oxygen therapy, suggesting that the oxygen intervention had broader beneficial effects on care processes. We found no change in care processes for children with diarrhoea and severe dehydration, a condition for which pulse oximetry and oxygen therapy have minimal role.

### How do changes in care practices for conditions correlate with observed changes (and lack of change) in clinical outcomes?

In previous analysis we reported the improved oxygen system intervention was associated with most significant clinical benefit for children with pneumonia (without detectable change in mortality for children with malaria or diarrhoea), and that most of this improvement was attributable to introduction of pulse oximetry [5].

In this analysis, we find similar improvements in QOC scores for children with severe pneumonia and children with severe malaria, and no change in QOC scores for children with severe diarrhoeal disease. However, the improvement in QOC for children with severe pneumonia appear to be primarily driven by oxygen-related care practices (i.e. pulse oximetry and oxygen therapy), whereas the improvement in QOC for children with malaria appears to be split evenly between oxygen-related and broader components of care. Furthermore, the prevalence of hypoxaemia among children with severe pneumonia was four-fold higher than among children with severe malaria or diarrhoea (45% versus 13 and 10% respectively) (S4 Table).

These results may suggest that the main driver of mortality benefit in the Nigeria Oxygen Implementation program was improved identification, monitoring and treatment of hypoxaemia (rather than broader improvements in quality of care), realising greatest impact in the population with highest hypoxaemia burden (children with pneumonia), despite the broader improvements in QOC detected for children with malaria.

Our results from Nigeria share similarities and differences with results from oxygen improvement programs in other low- and middle-income countries (LMICs) [3, 6]. In Papua New Guinea (PNG), Duke et al suggested that the most probable explanation for the observed 35% mortality reduction for children admitted to hospital with pneumonia is "the improved system and better quality of care that accompanies such a system" [3, 6]. Following this

rationale, subsequent work in Laos and remote clinics in the PNG highlands introduced improved oxygen systems alongside other clinical education and quality improvement activities—not only addressing pneumonia and oxygen therapy, but also many other important aspects of hospital care for children [6, 8]. These programs reported substantial mortality benefits for children with pneumonia, and the PNG program also documented substantial benefits for children overall [6, 8].

## How does a multi-faceted oxygen systems intervention affect broader care practices?

There are various ways that an oxygen improvement program could influence broader care practices, particularly relating to assessment and monitoring.

Qualitative data from our program in Nigeria suggested that nurses came to appreciate oximetry as a valuable tool for identifying and monitoring severely ill patients, saying it made their work easier and reduced stress [19]. This could conceivably lead to improvements in vital signs assessment and monitoring more broadly. However, while we observed clear improvements in the way oxygen was used as pulse oximetry practices improved, we did not observe broader effects on the assessment of other clinical signs or patient monitoring more broadly. This may be explained by relatively high baseline rates vital signs documentation, both on admission (>95% had vital signs documented) and over subsequent days (>85% had vital signs documented at least 3 times per day) (S5 Table). We did observe some improvement in vital signs monitoring during admission for children with severe pneumonia and malaria, but not for those with diarrhoea.

Pulse oximetry is the best way to detect hypoxaemia, with far greater sensitivity and specificity than any combination of clinical signs [16, 39, 40]. This could therefore assist healthcare workers in diagnosis, particularly for severe pneumonia [41]. However, we observed improved diagnosis scores for severe malaria, with no change in diagnosis scores for severe pneumonia or diarrhoea. This is despite relatively good diagnostic classification for children with severe malaria at baseline, but clear room for improvement for diagnostic classification of children with pneumonia (only one-third were identified as having severe pneumonia) (S5 Table). This finding of poor diagnostic classification of pneumonia is consistent with other reports from LMICs [42, 43] and is particularly concerning because accurate diagnosis correlates with correct treatment and survival [21].

In Laos, authors observed higher treatment completion rates for children with pneumonia, suggesting that oxygen therapy facilitated other aspects of treatment completion and prevented discharge against medical advice [6]. This may fit with anecdotal reports from healthcare workers that having pulse oximetry and an easy-to-use oxygen system makes work easier, but we had not necessarily expected our oxygen systems improvements to strongly influence broader treatment practices.

In fact, we observed improved treatment scores for children with severe pneumonia and malaria—over and above the improvements due to oxygen therapy. The changes in treatment for children with pneumonia were small, driven by improvements in fluid therapy. Notably, the proportion of children with severe pneumonia who received appropriate antibiotics remained very low even in the full oxygen system period (5.8%), largely due to the prescribing of broad-spectrum cephalosporins as first-line therapy (S5 Table).

The improvement in treatment for children with severe malaria was more substantial, driven by improvements in antimalarial prescribing (from 64% to 87%), with relatively good practices relating to anaemia (>98% received appropriate therapy) and fluids (~70% appropriate) across all time periods. With malaria recognition being relatively good in this

malaria-endemic region [21], pulse oximetry use may have formed part of a more confident or thorough assessment process, thereby contributing to general improvements in diagnosis and treatment [21]. But we are aware of malaria-related training that occurred in some participating facilities during the study period, and these may also have contributed to the observed change.

## Quality of care, context and emergence

We have presented data on care processes from a particular sub-Saharan African context and explored this in relation to a complex intervention. Quantitative quality of care scores have been shown to correlate with clinical outcomes [35]. However, quality of care scores can only measure some elements of quality of care, omitting important factors such as the relationships between, and attitudes of, healthcare workers, other staff, and patients [36, 44, 45]. Clinical outcomes emerge from the interaction between different care practices, particular interventions, and broader care structures and social contexts. Our findings do help explain how the Nigeria Oxygen Implementation program worked, particularly how improvements to hospital oxygen systems may influence broader aspects of care. However, they also raise many more questions for us, and for consideration in other contexts: how do oxygen-related care practices interact with other care practices: what degree of change in practice is required to translate into clinical benefits; how would this differ in different contexts or with interventions that addressed clinical care more holistically?

## Limitations

Our use of a stepped-wedge design was useful for evaluating quantitative outcome changes and was flexible enough to embed implementation learnings. However, it does not remove all confounding—particularly relating to time. First, are the major economic and social changes in Nigeria that occurred during the intervention, including change in government, economic recession and disruptions due to industrial action in state hospitals leading to hospital closures [46]. Delayed salary payment and economic hardship may have led to the demotivation of health facility staff, potentially altering or reducing the interventions impact on broader quality improvement [47]. Second, malaria awareness and training activities occurred in many hospitals during the study period as part of a national push for better diagnosis and treatment for severe malaria [48]. While we used a pre-specified analysis method, both positive and negative findings should be interpreted with caution due to the risk of chance findings.

Similar to others conducting facility-based implementation research, we extracted data from clinical documents [49]. We minimised the amount of missing data with the use of dedicated research nurses, with our audit suggesting that documentation practices overall were very good [22]. With the exception of standardised case definitions, we did not impose additional criteria on healthcare workers regarding their documentation. We assumed signs were not present if they were not documented, which could result in diagnostic misclassification or missed signs of severity—generally influencing results towards the null. We simplified the original scoring plan for antibiotics and antimalarials to omit requirements for correct dosing, due to inconsistencies with units recorded—generally increasing treatment scores.

## Practical implications

This study was part of a pragmatic field trial, which aimed to better understand how to improve oxygen access to children in Nigeria and other LMICs. Our findings provide practical lessons for those who seek to improve hospital oxygen systems and improve paediatric quality of care more broadly:

1. Improvement in hospital oxygen systems must be approached systematically. Efforts to build oxygen systems integrated with broader efforts to improve healthcare systems (e.g. equipment maintenance, emergency triage, case management, clinical documentation) may deliver greater improvements in the quality of care provided to patients—and better clinical outcomes. Oxygen is potentially a concrete focus on which broader improvement aims can be realised, if leveraged strategically.

2. Pulse oximetry is the key driver towards improved oxygen access. Together with improved oxygen delivery systems, pulse oximetry may drive broader improvements in the assessment and treatment of severely ill children and should be prioritised as the "fifth vital sign" for all children admitted to hospital.

These findings also highlight key areas for future practice and learning in the care of severely unwell children, including opportunities to: improve pneumonia assessment, diagnosis, and antibiotic prescribing; and further evaluate care processes (e.g. timeliness of treatment delivery) not measured in this study that may underpin a mortality benefit.

## Conclusion

We found that improvements in hospital oxygen systems were associated with higher quality of care scores for children admitted with severe pneumonia and severe malaria, in 12 Nigerian hospitals. These improvements were attributable to better use of pulse oximetry and oxygen as well as broader improvements in clinical care. We found no evidence of negative distortions in care practices resulting from oxygen systems interventions. Oxygen improvement programs may realise greater effect if oxygen improvement efforts are integrated alongside more comprehensive efforts to improve quality of care.

## Supporting information

**S1 Fig. Characteristics of the Nigeria Oxygen Implementation project.**
(DOCX)

**S1 File. Characteristics of 12 secondary-level hospitals in southwest Nigeria: Paediatric and neonatal wards (adapted from baseline needs assessment [26].**
(DOCX)

**S1 Table. Case definitions of severe pneumonia, severe malaria or diarrhoea with severe dehydration.**
(DOCX)

**S2 Table. Quality of care scores for severe pneumonia, severe malaria and diarrhoea with severe dehydration.**
(DOCX)

**S3 Table. Quality of care score restricted and extended analysis.**
(DOCX)

**S4 Table. Population characteristics of participants by study period and illness.**
(DOCX)

**S5 Table. Quality of care score feature and component sub-analysis.**
(DOCX)

## Acknowledgments

We thank the participating hospitals and their clinical, technical, and managerial staff. We thank all members of the Nigeria Oxygen Implementation team; representatives from the Federal Ministry of Health; representatives from the State Ministry of Health and hospital management boards in Oyo, Ondo, Ogun, and Osun states; and support staff at the Centre for International Child Health in Melbourne, Australia. We thank the study nurses Abiala Temitope, Adejubu Oyinlade, Adeleke Taiwo, Adesola Timilehin, Ajani Oluwatobi, Akamo Temitope, Akinrende Eniola, Alawode Juilet, Arubuolawe Elizabeth, Bakare Oluwabunmi Roseline, Eleyinmi Ayorinde Joseph, Kehinde Ayanfe, Mojoyinola Maria Oghenetekime, Ogbor Susan, Ogunkinle Alaba Segun, Ogunola Mayowa, Olaniyan Ibukun Abrouse and Sijuade Oluwaseun.

## Author Contributions

**Conceptualization:** Hamish R. Graham, Trevor Duke, Adegoke G. Falade.

**Data curation:** Hamish R. Graham, Ayobami A. Bakare, Oladapo B. Oyewole, Adegoke G. Falade.

**Formal analysis:** Hamish R. Graham, Jaclyn Maher, Cattram D. Nguyen.

**Funding acquisition:** Hamish R. Graham, Rasa Izadnegahdar, Trevor Duke, Adegoke G. Falade.

**Investigation:** Hamish R. Graham, Jaclyn Maher.

**Methodology:** Hamish R. Graham, Jaclyn Maher, Ayobami A. Bakare, Cattram D. Nguyen, Amy Gray, Trevor Duke, Adegoke G. Falade.

**Project administration:** Hamish R. Graham, Ayobami A. Bakare, Adejumoke I. Ayede, Adegoke G. Falade.

**Resources:** Adegoke G. Falade.

**Supervision:** Hamish R. Graham, Trevor Duke, Adegoke G. Falade.

**Visualization:** Hamish R. Graham, Jaclyn Maher.

**Writing – original draft:** Hamish R. Graham, Jaclyn Maher.

**Writing – review & editing:** Hamish R. Graham, Jaclyn Maher, Ayobami A. Bakare, Cattram D. Nguyen, Adejumoke I. Ayede, Oladapo B. Oyewole, Amy Gray, Rasa Izadnegahdar, Trevor Duke, Adegoke G. Falade.

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
