## [Decision Letter · Decision Letter 0]

1 Jun 2021

PONE-D-21-12894

Oxygen systems and quality of care for children with pneumonia, malaria and diarrhoea: analysis of a stepped-wedge trial in Nigeria

PLOS ONE

Dear Dr. Maher,

Thank you for submitting your manuscript to PLOS ONE. After careful consideration, we feel that it has merit but does not fully meet PLOS ONE’s publication criteria as it currently stands. Therefore, we invite you to submit a revised version of the manuscript that addresses the points raised during the review process.

For this initial peer review cycle, two content specific expert reviewers have been able to contribute their feedback on the submission, as detailed below. Generally their sentiments are positive, with just a few relatively minor suggested amendments proposed.

We look forward to receiving your revised manuscript.

Kind regards,

Shane Patman, PhD

Academic Editor

PLOS ONE

Journal Requirements:

In the ethics statement in the manuscript and in the online submission form, please provide additional information about the patient records used in your retrospective study, including: a) whether all data were fully anonymized before you accessed them; b) the date range (month and year) during which patients' medical records were accessed; c) the date range (month and year) during which patients whose medical records were selected for this study sought treatment. If the ethics committee waived the need for informed consent, or patients provided informed written consent to have data from their medical records used in research, please include this information.

Your ethics statement should only appear in the Methods section of your manuscript. If your ethics statement is written in any section besides the Methods, please delete it from any other section.

Please include captions for your Supporting Information files at the end of your manuscript, and update any in-text citations to match accordingly. Please see our Supporting Information guidelines for more information: http://journals.plos.org/plosone/s/supporting-information.

Reviewers' comments:

Reviewer's Responses to Questions

**Comments to the Author**

1. Is the manuscript technically sound, and do the data support the conclusions?

Reviewer #1: Partly

Reviewer #2: Yes

2. Has the statistical analysis been performed appropriately and rigorously? 

Reviewer #1: I Don't Know

Reviewer #2: Yes

3. Have the authors made all data underlying the findings in their manuscript fully available?

Reviewer #1: Yes

Reviewer #2: Yes

4. Is the manuscript presented in an intelligible fashion and written in standard English?

Reviewer #1: Yes

Reviewer #2: Yes

5. Review Comments to the Author

Reviewer #1: Authors have analyzed the following idea: To evaluate the effect of improved hospital oxygen systems on quality of

care (QOC) for 24 children with severe pneumonia, severe malaria, and diarrhea with severe dehydration.

Good article in general

Clear results with interesting clinical application

Reviewer #2: This is an important topic as there are major potential improvements in quality of care that could be made if the right vehicles for doing so were identified. This well-designed study has an adequate pre-intervention period and provides a relatively straightforward approach to assessing pre/post changes in quality of care in domains including and not including the directly provided intervention. The manuscript is clearly written.

It would be helpful in the initial study description to include how oxygen systems were to be continued and maintained once introduced and following the study, and any ongoing in-service training.

This may be in the other publications, but it would be useful to mention here whether any standardised admission clerking or monitoring forms were introduced as part of the intervention.

Why was assessment of nutritional status included in the QOC scores? Besides being a predominant predictor of outcome, it would make a difference in treatments in all three syndromes with respect to feeds and appropriate IV fluid therapy. The reported prevalence of malnutrition is very low for this kind of setting – how many actually had oedema and weight-for-height or MUAC assessed and recorded at admission?

It would be helpful to clarify in the text if the diagnoses of severe pneumonia, severe malaria or diarrhoea with severe dehydration were based on the clinician’s recorded diagnosis or on the presence of syndrome-defining clinical signs (page 6). Was malaria diagnosis based on slides or RDTs rather than clinical impression? Since diagnostic classification was reported to be ‘poor’, a valuable sensitivity analysis would be to restrict to patients meeting WHO syndromic criteria.

Was diagnostic classification assessed for severe malaria and diarrhoea with severe dehydration, and signs adequately recorded to assess this? If so, please give these data, if not, please discuss.

In relation to the role of oximetry and oxygen in diarrhoea and severe dehydration, it would be helpful to readers to give figures for hypoxia and shock by each of the three syndomes.

A common feature in governmental hospitals is high staff rotational turnover. Please add how frequent this was and measures taken to overcome it.

Was the training level of the clinicians assessing children and assigning treatments recorded? If so, could this be analysed?

6. PLOS authors have the option to publish the peer review history of their article (what does this mean?). If published, this will include your full peer review and any attached files.

Reviewer #1: No

Reviewer #2: **Yes: **James A Berkley

---

## [Author Response · Author response to Decision Letter 0]

15 Jun 2021

Dear PLOS ONE Reviewers

We thank you for comments and appreciate your time and thoughts. Below we detail our amended changes and response to your queries.

Please see the tracked changes document for our attention to formatting, particularly including changes to font size, figure and supporting file labelling.

2. In the ethics statement in the manuscript and in the online submission form, please provide additional information about the patient records used in your retrospective study

We have addressed the following in our ethics statement:

Whether all data were fully anonymized before you accessed them; yes, all data were anonymised, patient’s names were never collected and only dates of birth were gathered for age calculations.

The date range (month and year) during which patients' medical records were accessed; prospective & retrospective data collection occurred between September 2015 and December 2017.

The date range (month and year) during which patients whose medical records were selected for this study sought treatment; The admission period was November 2013 to October 2017.

If the ethics committee waived the need for informed consent, or patients provided informed written consent to have data from their medical records used in research, please include this information. The ethics committees did not require us to have individual patient consent.

Please see the tracked changes document for attention to this.

4. Please include captions for your Supporting Information files at the end of your manuscript, and update any in-text citations to match accordingly. 

Please see the tracked changes document for attention to this.

5. Please review your reference list to ensure that it is complete and correct. If you have cited papers that have been retracted, please include the rationale for doing so in the manuscript text, or remove these references and replace them with relevant current references. 

Nil changes were required.

Further queries/comments

- It would be helpful in the initial study description to include how oxygen systems were to be continued and maintained once introduced and following the study, and any ongoing in-service training.

All hospitals were provided with clinical and maintenance support, including three-monthly site visits for the first 18 months. Support was gradually withdrawn to encourage full site self-sustainability by the end of 2019, with formal handover and project finale. Our team remained available for remote advice and assistance on an ongoing basis, including for repairs and replacement of equipment. This information is published elsewhere but we have amended our study design description to be clearer.

- This may be in the other publications, but it would be useful to mention here whether any standardised admission clerking or monitoring forms were introduced as part of the intervention.

We did not introduce any standardised admission clerking or monitoring forms and relied on the same type of source documents throughout all phases of the study. Clerking doctors typically wrote out 1-2 pages of admission notes by hand, and nurses used a variety of monitoring forms to document vital signs. We provided education on vital signs monitoring, but all data was extracted from the routine clinical records by project study nurses using our case report forms. We have clarified this in our procedures section.

- Why was assessment of nutritional status not included in the QOC scores? Besides being a predominant predictor of outcome, it would make a difference in treatments in all three syndromes with respect to feeds and appropriate IV fluid therapy. The reported prevalence of malnutrition is very low for this kind of setting – how many actually had oedema and weight-for-height or MUAC assessed and recorded at admission?

Malnutrition was challenging for us to assess, and the best we could do was base it on healthcare worker documentation of MUAC, weight for height, or oedema meeting criteria for severe malnutrition (kwashiakor or marasmus) - and these assessments were generally only done if malnutrition was suspected (i.e. not a routine part of assessment). So, this is likely an underestimate of the true incidence of malnutrition and would not capture moderate acute malnutrition or stunting. This study was conducted in south-west Nigeria which is a region with lower rates of malnutrition compared to other regions of Nigeria (see image below from Amare 2018 paper in Food and Nutrition Bulletin). We did include specifications on fluid rate and type of fluid (e.g. Resomal vs ORS) in calculating the quality of care scores - but this made negligible difference to the scores. 

- It would be helpful to clarify in the text if the diagnoses of severe pneumonia, severe malaria or diarrhoea with severe dehydration were based on the clinician’s recorded diagnosis or on the presence of syndrome-defining clinical signs (page 6). Was malaria diagnosis based on slides or RDTs rather than clinical impression? Since diagnostic classification was reported to be ‘poor’, a valuable sensitivity analysis would be to restrict to patients meeting WHO syndromic criteria.

We used diagnostic classifications based on the presence of syndrome-defining clinical signs (not clerking clinician’s recorded diagnosis) to define the severe malaria, severe pneumonia and diarrhoea with severe dehydration cohorts. We have edited the text to make this clearer in our outcomes section.

- Was diagnostic classification assessed for severe malaria and diarrhoea with severe dehydration, and signs adequately recorded to assess this? If so, please give these data, if not, please discuss.

We assessed completeness of the clerking clinician’s diagnosis and recognition of severity for each of the diseases of interest. You can see the detailed breakdown of each of these score components in supplemental material (S5 Table. Quality of care score feature and component sub-analysis).

- In relation to the role of oximetry and oxygen in diarrhoea and severe dehydration, it would be helpful to readers to give figures for hypoxia and shock by each of the three syndromes.

Detailed information is contained in supplemental material (see S4 Table. Population characteristics of participants, by study period and illness).

- A common feature in governmental hospitals is high staff rotational turnover. Please add how frequent this was and measures taken to overcome it. Was the training level of the clinicians assessing children and assigning treatments recorded? If so, could this be analysed?

Yes, high staff rotational turnover was common, particularly in the government hospitals. In addition, there were periods of industrial action and closure of services, stemming from long periods of unpaid salaries. Our training and supervision methods encouraged local retraining using our educational materials, and this was reportedly typically done through small group in-service training at the point of care as new staff rotated onto wards (but we did not evaluate this specifically). Our training activities (and evaluation of HCW knowledge/skills) focussed on pulse oximetry and oxygen and did not extend to broader aspects of diagnosis or management of childhood conditions (unlike similar work in PNG which was much more holistic). We did not record individual patient level data on the cadre of treating clinician. In general, these were junior doctors (family medicine residents or house officers) without extensive paediatric experience - see supplemental materials for more information on facility staffing (S1 File - Characteristics of 12 Secondary-level hospitals in southwest Nigeria: paediatric and neonatal wards (adapted from baseline needs assessment).

We thank you for your time and look forward to hearing about the next steps.

Warm regards,

Jaclyn Maher - on behalf of the team

---

## [Decision Letter · Decision Letter 1]

23 Jun 2021

Oxygen systems and quality of care for children with pneumonia, malaria and diarrhoea: analysis of a stepped-wedge trial in Nigeria

PONE-D-21-12894R1

Dear Dr. Maher,

We’re pleased to inform you that your manuscript has been judged scientifically suitable for publication and will be formally accepted for publication once it meets all outstanding technical requirements.

Kind regards,

Shane Patman, PhD

Academic Editor

PLOS ONE

Additional Editor Comments (optional):

Reviewers' comments:

Reviewer's Responses to Questions

**Comments to the Author**

1. If the authors have adequately addressed your comments raised in a previous round of review and you feel that this manuscript is now acceptable for publication, you may indicate that here to bypass the “Comments to the Author” section, enter your conflict of interest statement in the “Confidential to Editor” section, and submit your "Accept" recommendation.

Reviewer #2: All comments have been addressed

2. Is the manuscript technically sound, and do the data support the conclusions?

Reviewer #2: Yes

3. Has the statistical analysis been performed appropriately and rigorously? 

Reviewer #2: Yes

4. Have the authors made all data underlying the findings in their manuscript fully available?

Reviewer #2: Yes

5. Is the manuscript presented in an intelligible fashion and written in standard English?

Reviewer #2: Yes

6. Review Comments to the Author

Reviewer #2: My comments have been addressed, continuing this sentence to meet the minimum word count for this box.

7. PLOS authors have the option to publish the peer review history of their article (what does this mean?). If published, this will include your full peer review and any attached files.

Reviewer #2: **Yes: **James A Berkley

---

## [Editor Report · Acceptance letter]

28 Jun 2021

PONE-D-21-12894R1 

Oxygen systems and quality of care for children with pneumonia, malaria and diarrhoea: analysis of a stepped-wedge trial in Nigeria 

Dear Dr. Maher:

I'm pleased to inform you that your manuscript has been deemed suitable for publication in PLOS ONE. Congratulations! Your manuscript is now with our production department. 

Kind regards, 

on behalf of

Assoc Prof Shane Patman 

Academic Editor

PLOS ONE